# Wet Bulb Globe Temperature and Recorded Occupational Injury Rates among Sugarcane Harvesters in Southwest Guatemala

**DOI:** 10.3390/ijerph17218195

**Published:** 2020-11-06

**Authors:** Miranda Dally, Jaime Butler-Dawson, Cecilia J. Sorensen, Mike Van Dyke, Katherine A. James, Lyndsay Krisher, Diana Jaramillo, Lee S. Newman

**Affiliations:** 1Center for Health, Work & Environment, Department of Environmental and Occupational Health, and Colorado Consortium for Climate Change & Health, Colorado School of Public Health, University of Colorado, Anschutz Medical Campus, 13001 E. 17th Pl., 3rd Floor, Mail Stop B119 HSC, Aurora, CO 80045, USA; jaime.butler-dawson@cuanschutz.edu (J.B.-D.); cecilia.sorensen@cuanschutz.edu (C.J.S.); mike.vandyke@cuanschutz.edu (M.V.D.); kathy.james@cuanschutz.edu (K.A.J.); lyndsay.krisher@cuanschutz.edu (L.K.); diana.jaramillo@cuanschutz.edu (D.J.); lee.newman@cuanschutz.edu (L.S.N.); 2Department of Medicine, School of Medicine, University of Colorado, Anschutz Medical Campus, 13001 E. 17th Pl., 3rd Floor, Mail Stop B119 HSC, Aurora, CO 80045, USA

**Keywords:** occupational injury, climate change, agricultural workers

## Abstract

As global temperatures continue to rise it is imperative to understand the adverse effects this will pose to workers laboring outdoors. The purpose of this study was to investigate the relationship between increases in wet bulb globe temperature (WBGT) and risk of occupational injury or dehydration among agricultural workers. We used data collected by an agribusiness in Southwest Guatemala over the course of four harvest seasons and Poisson generalized linear modelling for this analysis. Our analyses suggest a 3% increase in recorded injury risk with each degree increase in daily average WBGT above 30 °C (95% CI: −6%, 14%). Additionally, these data suggest that the relationship between WBGT and injury risk is non-linear with an additional 4% acceleration in risk for every degree increase in WBGT above 30 °C (95% CI: 0%, 8%). No relationship was found between daily average WBGT and risk of dehydration. Our results indicate that agricultural workers are at an increased risk of occupational injury in humid and hot environments and that businesses need to plan and adapt to increasing global temperatures by implementing and evaluating effective occupational safety and health programs to protect the health, safety, and well-being of their workers.

## 1. Introduction

Temperatures in Central America have steadily increased 0.1 °C per decade since 1960 and as of 2014 are expected to continue to increase upwards of an additional 4 °C by the year 2100 [1]. Increasing temperatures, as well as increasing frequency and intensity of heat waves, are expected to create an additional burden on workers and the business enterprise [2]. It is estimated that by 2050 there could be an annual loss of 880 million labor hours and USD 44 billion in lost wages [3], with outdoor workers, especially those conducting intense manual labor, expected to be those most impacted [4]. Already, crop production is among the most hazardous jobs in agriculture with an incidence rate of non-fatal occupational injury and illness of 5.9 per 100 workers in the United States [5], and likely higher in Central America due to lack of national regulations and centralized reporting [6]. It is estimated that in the United States, crop workers die from heat stroke at a rate nearly 20 times greater than all civilian workers [7]. With nearly 866 million workers worldwide officially employed in the agricultural sector [8], it is imperative that we understand the impacts of rising temperatures as well as extreme temperature events on the ability of workers to maintain their health and livelihood.

Ambient heat exposure is a particular hazard for agricultural workers because it adds to internal heat production while strenuous work is carried out [9]. The additive effects of high ambient temperature and increases in metabolic heat production from strenuous work can increase core body temperature, resulting in a spectrum of heat illnesses including, dehydration, heat edema, heat syncope, heat exhaustion, heat stroke, and death [10]. Increased ambient temperature is associated with increased risk of mortality, particularly among vulnerable populations, due to heat illness and the effects of heat on underlying medical conditions [11,12]. In addition to heat specific injury and illness, heat exposure has been associated with an increased risk of workplace injuries [13]. Heat exposure and early heat illness can manifest as central nervous system dysfunction that affects coordination, fine motor skills and judgement and similarly result in fatigue and dizziness [10,14] which may exacerbate injury risk, including falls, wounds, lacerations, and amputations [15].

Dehydration is a particular area of concern for workers laboring in hot environments as it can increase the risk for heat-related illness, rhabdomyolysis, and exercise-associated hyponatremia [16]. Among sugarcane harvesters, dehydration has been linked to Chronic Kidney Disease of Unknown Origin [17,18,19], an epidemic which has resulted in tens of thousands of deaths among workers in Latin America [20]. Additionally, it has been hypothesized that fatigue, muscle cramps, and dehydration may be contributory physiological factors that act as precursors to work injuries in hot weather [21].

Currently, very few studies have examined the impact of increasing temperatures on worker injury rates. One study examining the impact of ambient temperatures on occupational injuries in Spain showed that nearly 3% of all occupational injuries, regardless of occupation, were attributed to conducting work in high temperatures [22]. In Guangzhou, China, it was estimated that the relative risk (RR) of occupational injury was 1.15 when working at 30 °C compared to 25 °C [23]. For young male workers in Australia, every 1 °C increase in daily maximum temperature resulted in the odds of injury increasing by 0.8% [24], while in Quebec, Canada, researchers found that the risk of occupational injury for manual agricultural workers increased 0.2% for each 1 °C increase in daily maximum temperature [13].

While these aggregated studies have provided valuable estimations, they are limited in their ability to understand industry-specific risks, specifically for those who will likely be impacted the most by increasing temperatures, such as agricultural workers. Additionally, many of these studies have been conducted in fairly mild climatic regions using measures of ambient temperature and do not provide insight into risk at temperatures regularly observed in Central America, where average wet bulb globe temperatures (WBGT) range from 25.8 to 31.9 °C during the working day [25]. The WBGT incorporates humidity, radiation, and windspeed along with ambient temperature making it a stronger measure of heat stress index than ambient temperature alone [26]. Furthermore, by examining aggregated data, these studies are limited in their ability to provide insight into the types of injuries that are likely to increase due to the changing climate, hindering their ability to provide a basis for targeted occupational safety and health interventions.

In this paper, we assess the relationship between WBGT and all company recorded occupational injuries among sugarcane harvesters at a sugarcane mill in Southwest Guatemala from 2014 to 2018. We then examine the association between WBGT and injuries specific to being cut by an agricultural tool, slips, trips, and falls. From 2016 to 2018, we examine how increases in WBGT impact the risk of heat-induced dehydration illness experienced on the job. We hypothesize that as WBGT increases so will the risk of all recorded occupational injury, with the strongest relationships hypothesized to be between WBGT and cuts, slips, trips, and fall injuries and heat-induced dehydration illness.

## 2. Materials and Methods

### 2.1. Sugarcane Harvesting

Manual sugarcane harvesting involves swinging a machete to cut the sugarcane stalk a few centimeters above ground level, followed by lifting, trimming, and stacking the cane [27]. Manual sugarcane harvesting is considered heavy to very heavy metabolic work with estimates showing 6.8 kCal burned per minute [25], with workers cutting approximately 6 tons of sugarcane per shift [27]. A typical shift lasts from 7:00 to 17:00 with three 20-min rest breaks and a 60-min lunch break. Workers work for six consecutive days before one rest day. A typical sugarcane harvest lasts from November to April.

In each harvest, there are approximately 4000 male manual sugarcane harvesters employed at the mill providing data for this analysis. Workers come from both the local communities around the mill as well as workers that migrate from higher altitude communities within Guatemala. Local workers live in their own homes during the harvest and commute to the fields each day, while migrant workers are housed within the mill and are transported to the fields each day via bus. The company enforces an early season acclimatization period, during which workers labor fewer hours and cut less sugarcane, typically lasting the first two weeks of the harvest. Since 2009, the sugarcane mill has focused on promoting hydration, electrolyte replacement, rest, and shade, based on of the U.S. Occupational Safety and Health Administration (OSHA) recommendations [28]. A full summary of the company’s health and safety practices for field workers has previously been reported [29]. Notably, all workers are provided with access to clean, chlorinated water that has been checked for coliform bacteria and metal contamination, as well as with electrolytes in the field. Workers are issued a personal, refillable 5-L container for water by the agribusiness. Mobile tanks of clean water are accessible at any time and are located at a central point in the field. Electrolyte solution is provided to the workers at the start of each workday [30].

### 2.2. Wet Bulb Globe Temerpature

Wet bulb globe temperature (WBGT) has been well tested and used in the development of suggested occupational work-rest cycles to prevent heat injury among outdoor workers [26]. Following methods we have previously described [27], we calculated the average daily WBGT (WBGT_mean_) and the maximum daily WBGT (WBGT_max_) during the hours of 7:00–17:00 using data from the El Balsamo weather station (14.28° N, 91.00° W, 280 m above sea level). The cutting groups rotate through numerous plantations, ranging from sea-level to 500 m, throughout the Department of Escuintla. The choice of the El Balsamo weather station was made since: (1) it lies along the central border of the field range; (2) is within approximately 21,600 m of the majority of fields; and (3) is at an altitude of 280 m, the average altitude found for all fields.

### 2.3. Data Sources

Occupational injury logs and daily productivity logs tracking the amount of sugarcane cut by each individual per day were collected and maintained by the sugarcane mill in Southwestern Guatemala and were provided in de-identified form to researchers at the University of Colorado. Occupational injuries that were reported by the worker to their supervisor overseeing field operations that day or the field nurse who was present are recorded on an incident log. Data for this analysis were available for November through April from the 2014–2015, 2015–2016, 2016–2017, and 2017–2018 harvests. We limited this analysis to only manual sugarcane harvesters.

Institutional review for the evaluation of these data was completed by the Colorado Multiple Institutional Review Board (COMIRB #18-0957). COMIRB determined that informed consent was not required for the evaluation of these deidentified data, since they had been previously collected for business and clinical purposes.

### 2.4. Variables of Interset

Annual injury rate was calculated following the methods of the Bureau of Labor Statistics [5]. We divided the total observed number of injuries by the number of years of observation. We divided this number by the average number of individuals who worked each day multiplied by 10 h a day, 6 days a week, for 23 weeks (average length of a harvest). The resulting value was then multiplied by a base of 138,000 representing 100 workers working 10 h a day, six days a week, for 23 weeks. The resulting value provided the estimated annual rate per 100 workers.

#### 2.4.1. Outcome Variables

The primary outcome of interest was the total number of recorded occupational injuries among sugarcane harvesters per day. This included injuries in the following categories: falls, hit by a falling object, slips, caught or stuck between an object, strains or sprains, exposure to extreme heat (non-ambient such as steam), exposure to electrical current, exposure to a harmful substance or radiation, chemical accidents, cut with an agricultural tool, vehicular accidents, bites from snakes or insects, agricultural incidents, or other. Secondary outcomes included the total number of recorded occupational injuries specific to cuts with an agricultural tool, falls, and slips given the hypothesized mechanism of association with heat exposure [15].

Information on dehydration was recorded starting with the 2016–2017 season. Prior to the 2016–2017 season, the company had not considered dehydration as a reportable incident. As an additional analysis, we examined the total number of confirmed dehydration cases per day for the 2016–2017 and 2017–2018 harvests. Dehydration was self-reported by the workers to the field nurses based on symptoms of headache, weakness, and nausea and was clinically confirmed by medical staff based on urinary specific gravity reading of >1.020 prior to being recorded.

#### 2.4.2. Primary Predictor Variable

The main predictor of interest was mean WBGT (WBGT_mean_) during work hours for the day. The WBGT_mean_ variable was centered at the average value over the course of the study (29.9 °C) and treated as continuous in all analyses.

#### 2.4.3. Control Variables

To account for varying work attendance per day, we calculated the total number of workers per day by assessing the provided productivity data. If an amount of tons cut was recorded for an individual on a given day that worker was considered to have worked that day and the worker was counted in the total population for that day. We divided the daily total population by 1000 and included it as a covariate to account for the varying number of at-risk workers on a given day. We averaged the recorded tons of cane cut over each individual recorded working for a day for a daily average sugarcane cut variable, to be used as a measure of intensity. We created an indicator variable for the first two weeks of the harvest to account for the acclimatization period where workers tend to work fewer hours cutting less sugarcane.

### 2.5. Statistical Analyses

#### 2.5.1. Functional form of the Relationship between WBGT and Occupational Injury

Previous studies examining the association between WBGT and health outcomes have suggested that the appropriate functional form may take on a U-shape or other non-linear relationship indicating that both extreme cold and extreme heat are risk factors for morbidity and mortality [31]. Given the WBGT ranges observed in Central America [25], extreme cold is not a concern currently in this population. Nevertheless, we assessed correlation plots between WBGT and reported count of injuries and fit a smoothed spline for each of the four harvests independently to understand if similar non-linear trends were present at the extreme temperatures experienced by our study population.

#### 2.5.2. Regression Modelling

As the outcome was count data, we considered a generalized linear model (GLM) with a Poisson distribution for the outcome and a log link function. We also considered a zero-inflated Poisson model to account for the potential of excess days where the recorded injury count was zero. To determine the appropriate assumption, we ran both models examining total recorded injury count on centered WBGT_mean_ while adjusting for total number of workers and harvest year. All harvests were included in the regression models. A Vuong test [32] was used to compare the models.

We confirmed our visual conclusion regarding the functional form of WBGT by running the GLM described above with the addition of a quadratic term for centered WBGT_mean_. We compared the models with and without this additional term using a likelihood ratio test.

To create our final model for analysis, we added the additional covariates of acclimatization period indicator (first 2 weeks of harvest vs. rest of harvest) and average daily tons cut to the appropriate model with the correct functional form for centered WBGT_mean_. We ran the multivariable models independently for the three outcomes of interest: total daily recorded occupational injuries, total daily recorded cut, slips, and fall injuries, and total daily confirmed dehydration cases. All statistical analyses were conducted using R version 3.6.1 [19].

#### 2.5.3. Sensitivity Analyses

To test the sensitivity of our conclusions to the choice of WBGT summary statistic, we re-ran the multivariable model for total daily recorded occupational injuries with centered WBGT_max_. To understand the stability of the estimates to selection of weather station, we re-ran the multivariable model for total daily recorded occupational injuries using data from the Cengicaña weather station (14.33° N, 91.05° W, 300 m above sea level). Data from Cengicaña were available only for the 2015–2016 harvest, so we limited this analysis to only injuries recorded during the 2015–2016 harvest.

## 3. Results

### 3.1. Occupational Injury Counts and Rates

There was a total of 201 recorded injuries from 2014–2018 over the course of 711 working days. There was a total of 72 confirmed dehydration cases during 339 working days from 2016–2018. There were approximately 2734 (SD: 627; Min: 3, Max: 4042) men cutting sugarcane on any given day from 2014–2018. The majority of the occupational injuries recorded were cuts by an agricultural tool, falls, or slips (N = 163; 81%), with a vast proportion of these from being cut by an agricultural tool (N = 111; 68%). For the years in which dehydration data were collected, it was recorded twice as often as any other occupational injury (Table 1).

The estimated annual total recorded injury rate was 1.84 per 100 workers for 2014–2018. The estimated annual rate for dehydration was 1.57 per 100 workers for 2016–2018. (Table 2). The average daily total recorded injury rate was 0.10 per 1000 workers (SD: 0.20). The average daily recorded injury rate due to cuts, falls, or slips was 0.08 per 1000 workers (SD: 0.18), and the average daily rate of recorded dehydration was 0.09 per 1000 workers (SD: 0.21) (Table 3).

### 3.2. Wet Bulb Globe Temperature

The average WBGT during working hours on days in which sugarcane cutting occurred during the 2014–2018 harvest seasons was 29.9 °C (SD: 1.6). Daily WBGT_mean_ was highly correlated with daily WBGT_max_ (r = 0.88). We observed a general trend of WBGT_mean_ becoming progressively hotter throughout the study period (*p*-value: <0.0001). For the 2014–2015 harvest, the average WBGT_mean_ was 28.9 °C (SD: 1.4); for the 2015–2016 harvest, the average WBGT_mean_ was 29.9 °C (SD: 1.3); for the 2016–2017 harvest, the average WBGT_mean_ was 30.7 °C (SD: 1.7); and for the 2017–2018 harvest, the WBGT_mean_ was 30.3 °C (SD: 1.2). Exploratory analysis suggested that mean daily average humidity increased over the same time course (Appendix A), while mean daily average ambient temperatures slightly decreased (Appendix A). Within each season we observed a general decrease in WBGT_mean_ from November to February and then an increase in temperatures from February through April (Figure 1). It was observed to be slightly hotter (0.3 °C; 95% CI: −0.04, 0.66; *p*-value 0.079) during the acclimatization phase of each harvest (30.2 °C) compared to the rest of the harvest (29.9 °C).

### 3.3. Association of WBGT with Occupational Injury

The relationship between WBGT_mean_ and daily total recorded occupational injury counts appeared to be quadratic for each of the four seasons independently (Figure 2) and a likelihood ratio test suggested the inclusion of the quadratic term provided marginally better fit (*p*-value: 0.050). The Vuong test suggested that the Poisson GLM with a log link function was sufficient (AIC corrected *p*-value: <0.001 for superiority of the GLM).

After adjusting for the number of workers, harvest year, average productivity, and whether it was during the acclimatization period, for every 1 °C increase in centered WBGT_mean_ above 30 °C, the expected mean count of daily recorded occupational injury increased by 3% (95% CI: −6%, 14%; *p*-value: 0.544) (Table 4), although this was not statistically significant. The quadratic term indicates that the relationship between centered WBGT_mean_ and daily total injury count accelerated by 4% (95% CI: 0%, 8%; *p*-value: 0.034) for every degree above 30 °C. The multiplicative increase in rate (acceleration) is demonstrated in Figure 3. For example, the expected daily rate of total injury in the 2014–2015 harvest season is 0.223 per 1000 workers when the temperature is 30 °C, the average amount of sugarcane cut is 6 tons, and it occurs after the first 2 weeks of the harvest. The expected rate increases to 0.239 under these same conditions if the temperature rises to 31 °C and further increases to 0.256 at 32 °C.

When examining only injuries related to being cut with an agricultural tool, slips, or falls similar relationships with centered WBGT_mean_ were observed (Table 4). Each degree increase in centered WBGT_mean_ beyond 30 °C increased the risk of cut, slips, or falls by 7% (95%CI: −4%, 20%; *p*-value: 0.224), although this was not statistically significant. Similarly, the quadratic term for centered WBGT_mean_ contributed to a 3% acceleration beyond 30 °C (95% CI: −1%, 7%; *p*-value: 0.151). For dehydration, no effects attributable to centered WBGT_mean_ were observed (Table 4).

Both total injury and cut, slip, and fall injuries saw a non-statistically significant increased risk in the acclimatization period (17% and 31%, respectively); however, the count of confirmed dehydration is expected to be 15% lower during the acclimatization period (95% CI: −87%, 336%; *p*-value: 0.835). The data suggest that the amount of sugarcane cut increased the risk of total injury and cut, slip, and fall injuries, but had no observable effect on dehydration.

Sensitivity analyses showed similar results when examining the relationship between daily recorded occupational injury and WBGT_max_ (Appendix A). Despite highly correlated observations between the two weather stations during the 2015–2016 harvest (r = 0.91; Appendix A), results were sensitive to the selection of weather station. Results using the El Balsamo weather station showed larger effect sizes and smaller standard errors than those using the Cengicaña weather station (Appendix A).

## 4. Discussion

This is the first known study that examines the risk of increasing WBGT on recorded occupational injury among agricultural workers in hot tropical climates. Our data suggest that occupational injury rates among agricultural workers increase with increasing temperatures, even among acclimatized workers. We established that the expected rate of increase is not linear, but rather it accelerates in a quadratic fashion when average temperatures exceed 30 °C. Additionally, we have shown that the highest risk period for occupational injury is within the first two weeks of starting strenuous physical labor.

Workers are often thought of as “climate canaries” [33], experiencing the effects of climate change at greater intensities and for longer durations than the general public, thus giving clues to how future climate scenarios will affect the general population. This is especially true for agricultural workers, who by the nature of their job, are unable to avoid the ambient environment while carrying out their work tasks. Although expert panels have established occupational adaptation strategies for extreme heat, many are not practical for the conditions where these and millions of other workers must make a livelihood. One such example is the recommended work-rest ratios for preventing heat stress. Under such a model, agricultural work, such as sugarcane harvesting, is recommended at 75% effort when WBGT is at or below 27.5 °C, with only up to 25% work recommended when WBGT exceeds 30.5 °C [34]. However, as Crowe et al. demonstrated, in Central America temperatures often rise above this level by 09:15 [25], and nearly 50% of the days in the present study had an average WBGT that surpassed 30.5 °C. With the understanding that manual harvesting of crops must be conducted regardless of temperature, it is imperative to determine the excess risk that workers assume when they labor under conditions of higher temperatures.

In this study, we established that the risk of recorded occupational injury increases 3% for each degree increase in average daily mean WBGT above 30 °C. This is in line with the study by Sheng and colleagues which estimated that risk of occupational injury, regardless of industry, increased 1.4% for each degree increase in daily maximum temperature and 1.7% for each degree increase in daily minimum temperature [35]. The average daily maximum temperature in the Sheng study was 32 °C with the average minimum of 24 °C. The differences in increased risk are unsurprising given agricultural workers have been found to have excess risk of experiencing occupational injury in light of increasing temperatures [36,37].

Although many studies have been conducted allowing for the non-linear relationship between heat exposure and occupational injuries, there are no studies that we are aware of that assess the acceleration of risk beyond a given high heat threshold. Our finding that risk accelerates by 4% for every degree beyond 30 °C implies that linear assumptions are invalid. Notably, in the present study, we were able to examine the relationship at higher WBGT than are typically assessed. With data points exceeding 34 °C, this is the first study we are aware of that provides more stable estimates at the higher range of observed average daily temperatures without relying on extrapolation. Our findings demonstrate that the relationship between heat and occupational injury differ at the extreme high temperatures and indicate that regionally specific guidelines should be implemented, as currently published estimates may not be valid for hot tropical climates.

Our data provide valuable insight into the timing of occupational injuries. Unsurprisingly, we found that the risk of injury was highest in the first two weeks of the harvest. Interestingly, far fewer dehydration injuries were recorded during this period despite there being no observed association with the average daily WBGT. Focusing on the two most recent harvests, dehydration was recorded at almost twice the rate as any other occupational injury. This is likely due, in part, to the increased focus of the agribusiness on water, electrolyte replacement, rest, and shade as part of their Total Worker Health ^®^ (TWH) response to the epidemic of Chronic Kidney Disease of Unknown Origin [29]. Arguably, the rates of recorded dehydration incidents are indicative of the increased efforts around educating workers on the importance of hydration and recognizing the signs of dehydration, indicating that the employer takes the issue seriously and has created an environment where such illnesses are encouraged to be reported. Without prior data it is hard to determine if the educational programs have decreased the rates of dehydration; however, these efforts demonstrate a successful implementation of an educational program at the business level, recognizing and addressing heat related illnesses. Such models support the notion that monitoring worker health is essential to the process of climate adaptation within the workplace.

Interestingly, we saw a marked reduction in the annual rate of recorded occupational injury per 100 workers over the course of the four harvest seasons. One potential explanation for this is the agribusiness’ increased investment in safety starting in 2015. With the hiring of a corporate level safety manager to oversee safety at all levels of the organization, there was a strengthening of the safety training program and an indication that the agribusiness made safety a priority, which is reflected in a strong safety culture. Despite this, the estimated annual rate of 1.84 injuries per 100 workers observed in this study is 50% fewer than those observed for crop harvesters in the United States where the annual rate is 5.2 per 100 workers [5]. This may be the consequence of severe underreporting of injuries by workers or under recording by the employer, which are not uncommon practices in the agricultural setting [38]. Despite a robust Safety Culture program at the agribusiness that encourages reporting, psychological safety to report an injury remains a barrier. Additionally, as wage is reliant on the ability of the worker to cut sugarcane, it is likely that a worker does not want to stop working to report a minor/moderate injury. Issues of underreporting likely lead to the attenuation of our results to the null. Before tailored OSH approaches in the workplace can be implemented, it is imperative that employers create a culture of safety that is encouraging of reporting without fear of retribution and conduct independent audits to ensure accurate recording.

While on average the trend of WBGT appeared to increase throughout the years, it was not monotonically increasing, suggesting that lower temperatures during the 2017–2018 harvest may have contributed, in part, to the observed reduction in annual rate of reported injury per 100 workers. Additionally, given that we aggregated at the workforce level, we were unable to account for individual factors which may increase or decrease an individual’s risk of injury. These factors include age, personal health, sleep quality, and behaviors such as smoking or alcohol use [39]. These same factors can mediate an individual’s physiological response to heat [40]. Coupled with the improvements to safety culture, this suggests that any OSH adaptation strategy to address increasing outdoor occupational heat exposure due to climate change should take a Total Worker Health approach, systematically addressing the contribution of multiple factors that may contribute to injury risk.

While this study provides valuable insight into the associations between average daily mean WBGT and recorded occupational injury among sugarcane harvesters the interpretation of our results is limited in a few ways. Our study leveraged data from a single agribusiness in a localized region, weakening the generalizability to all agricultural workers. The reliance on worker reporting and employer recording of data regarding both incidence and nature of the injury data likely did not capture all injuries leading to potential biasing of our results. Dehydration data were collected over only two years and only for those individuals who recognized the signs and symptoms, thus reducing our ability to detect an effect of WBGT on dehydration incidents. Measurements of WBGT were calculated using data from a single weather station. While our selection was based on proximity to the majority of fields where the sugarcane harvesters were working, we were unable to geographically locate each injury site. The choice of a single weather station for WBGT exposure measurement likely led to misclassification bias. A sensitivity analysis conducted on a subset of the data demonstrated that our observed results were sensitive to the selection of weather station. Additionally, our analytical approach did not allow us to account for the impact of heatwaves. Exploratory analysis suggested that increased trends in average daily WBGT may be attributable to humidity, suggesting that future research should examine the contribution of humidity to occupational injuries and dehydration.

## 5. Conclusions

This study provides valuable insight into how increases in average daily WBGT, especially above 30 °C, are impacting the health of agricultural workers in hot tropical regions. It establishes that workers exposed to increasing heat are more likely to experience and report any occupational injury, not just heat stress and dehydration which are often the only outcomes examined, and that this relationship is non-linear but rather accelerates as temperatures increase. Given the forecasts for the globe, it is imperative to understand the impacts that increasing temperatures will have on working populations. Acknowledging that agricultural workers will be at an increased risk of occupational injury will allow employees, employers, and regulatory agencies to plan and adapt by implementing and evaluating effective OSH programs to protect the health and safety of the agricultural workforce.

## Figures and Tables

**Figure 1 ijerph-17-08195-f001:**
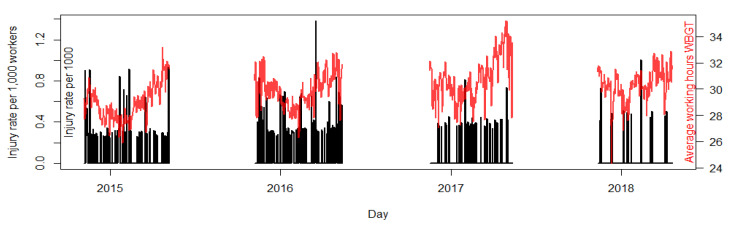
Daily total recorded occupational injury rate per 1000 Southwestern Guatemalan sugarcane harvesters during the 2014–2015 to 2017–2018 harvest seasons along with daily WBGT_mean_ temperatures. Harvest seasons run from November to April and are indicated by the annual label on the plot.

**Figure 2 ijerph-17-08195-f002:**
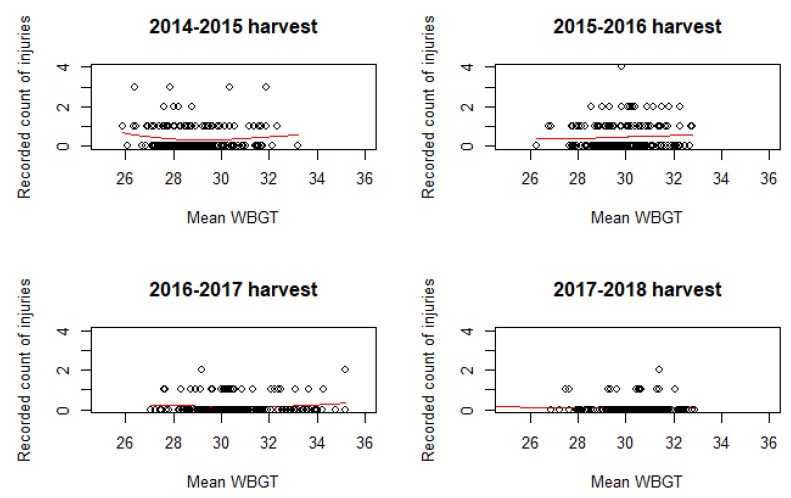
Correlation between daily total recorded occupational injuries reported by Southwestern Guatemalan sugarcane harvesters with daily WBGT_mean_ temperatures. Fitted smooth spline modeled independently for 2014–2015 through 2017–2018 harvests demonstrates the non-linear relationship between occupational injury and WBGT_mean_.

**Figure 3 ijerph-17-08195-f003:**
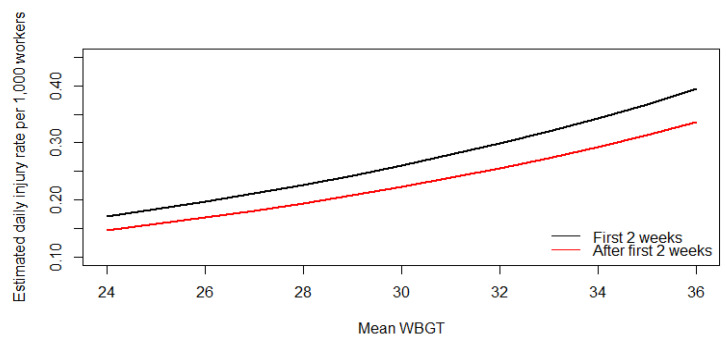
Estimated daily total recorded occupational injury rate per 1000 sugarcane harvesters with daily WBGT_mean_ temperatures for the 2014–2015 harvest with the assumption of an average of 6 tons of sugarcane harvested per day.

**Table 1 ijerph-17-08195-t001:** Summary ^1^ of recorded occupational injuries among sugarcane harvesters in Southwestern Guatemala—2014–2018.

Injury Type	2014–2015	2015–2016	2016–2017	2017–2018	Overall
Total injuries	67	87	33	14	201
Cuts, falls, or slips	64 (96%)	68 (78%)	22 (67%)	9 (64%)	163 (81%)
Cuts	38 (58%)	48 (55%)	16 (49%)	9 (64%)	111 (55%)
Falls	1 (2%)	9 (10%)	4 (12%)	0 (0%)	14 (6%)
Slips	25 (37%)	11 (13%)	2 (6%)	0 (0%)	38 (19%)
All other	3 (4%)	19 (22%)	11 (33%)	5 (36%)	38 (19%)
Dehydration ^2^	–	–	55	17	72

^1^ Summary data represented as number of recorded injuries and percent of total recorded injuries. ^2^ Data only collected for 2016–2017 and 2017–2018 harvests. Dehydration data not included in total injury count.

**Table 2 ijerph-17-08195-t002:** Estimated annual rate of recorded injury and dehydration per 100 sugarcane harvesters in Southwestern Guatemala, 2014–2018.

Annual Rate Per 100 Workers	2014–2015	2015–2016	2016–2017	2017–2018	Overall
Total injury rate	2.02	2.94	1.33	0.68	1.84
Cuts, falls, or slips rate	1.93	2.30	0.88	0.44	1.49
Dehydration rate ^1^	–	–	2.21	0.82	1.57
Average # of workers	3319	2962	2491	2066	2734

^1^ Data only collected for 2016–2017 and 2017–2018 harvests.

**Table 3 ijerph-17-08195-t003:** Estimated average daily rate of recorded injury and dehydration per 1000 sugarcane harvesters in Southwestern Guatemala, 2014–2018. Presented as mean (SD).

Daily Rate Per 1000 Workers	2014–2015	2015–2016	2016–2017	2017–2018	Overall
Total injury rate	0.11 (0.20)	0.16 (0.23)	0.07 (0.17)	0.04 (0.15)	0.10 (0.20)
Cuts, falls, or slips rate	0.11 (0.20)	0.12 (0.21)	0.05 (0.14)	0.03 (0.12)	0.08 (0.18)
Dehydration rate ^1^	–	–	0.12 (0.25) ^2^	0.05 (0.17)	0.09 (0.21)

^1^ Data only collected for 2016–2017 and 2017–2018 harvests. ^2^ Single day with a rate of 250 (1 observation on a day with 4 workers) removed from average presented in table. Daily rate per 1000 workers with this day included was 1.15 (18.68).

**Table 4 ijerph-17-08195-t004:** Estimated Risk Ratios (RR) for reported occupational injuries and confirmed dehydration cases among sugarcane harvesters in Southwestern Guatemala—2014–2018. All models adjusted for total number of workers and harvest season.

	All Injuries ^1^		Cut, Slips, or Falls		Dehydration ^2^	
	RR (95% CI)	*p*-value	RR (95% CI)	*p*-value	RR (95% CI)	*p*-value
Centered ^3^ WBGT_mean_	1.03 (0.94, 1.14)	0.544	1.07 (0.96, 1.20)	0.224	1.01 (0.80, 1.39)	0.927
Centered WBGT_mean_ ^2^	1.04 (1.00, 1.08)	0.034	1.03 (0.99, 1.07)	0.151	1.01 (0.92, 1.08)	0.769
Average daily tons cut	1.09 (0.89, 1.32)	0.404	1.17 (0.94, 1.45)	0.160	1.00 (0.61, 1.62)	0.999
Acclimatization period ^4^	1.17 (0.64, 1.99)	0.584	1.31 (0.68, 2.29)	0.385	0.85 (0.13, 3.36)	0.835

^1^ All recorded occupational injuries to include: falls, hit by a falling object, slips, caught or stuck between an object, strains or sprains, exposure to extreme heat (non-ambient such as steam), exposure to electrical current, exposure to a harmful substance or radiation, chemical accidents, cut with an agricultural tool, vehicular accidents, bites from snakes or insects, agricultural incidents, or other. ^2^ Data only collected for 2016–2017 and 2017–2018 harvests. ^3^ WBGT_mean_ was centered on 30 °C. ^4^ Acclimatization period occurred during the first 2 weeks of every harvest.

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
