# Peer review of "Wet Bulb Globe Temperature and Recorded Occupational Injury Rates among Sugarcane Harvesters in Southwest Guatemala"

_ijerph, 2020, doi:10.3390/ijerph17218195_

Round 1
Reviewer 1 Report
Suggestions for authors
Major comments
- This paper is interesting and a worthwhile contribution to the discussion of the impact of heat on workers, in the context of a warming climate.
- The Introduction needs to make a clearer rationale for including analysis of the impact of WBGT on the risk of dehydration in this study. There is also limited data available for analysis of this issue. One option would be to include this in a separate study once more data has been collected. This is a very important issue too.
- The limitations section of the discussion needs to address the issues of self-reported data (potential bias), use of a single weather station and limited data about dehydration.
- I notice that there is little discussion about ethics considerations of this study, other than to say that the research had been deemed ‘non-human subject research’. Was any consent required from workers? Were they aware that data was being collected about them?
Minor comments
- Introduction
A more in-depth rationale for including an assessment of the relationship between WBGT and risk of heat-induced dehydration illness in the study aim is needed. This appears somewhat ‘out of the blue’ in the final paragraph of the introduction.
Line 31/32: is any more recent data available? Reference 1 is dated 2014. This should be mentioned.
- Material and methods
- Sugarcane harvesting
Line 91/92: what does collaborating for this analysis mean? Did all these workers consent to providing their data for the research? This should be stated more clearly.
Line 100: Were workers provided with clean water or did they have to walk some distance to access this? Was any time penalty associated with accessing the water and electrolytes? This some elaboration.
- Data Sources
Line 120/121: Why was this research deemed ‘non-human subjects research’?
Line118: does this mean data was available for just November to April of each financial year? It would be good to state this.
2.4.1. Outcome variables
Line 138: What was the rationale for using information about dehydration starting with the 2016/2017 season, rather than the 2014/2015 season? Was the 2014/2015 and 2015/2016 data less complete? This needs some explanation.
Line 142-144: Please explain the rationale for this.
- Results
Line 187: grammar, should be ‘there were’…
Line188: grammar, should be ‘there were’…
Line 193: needs rewording. Do you mean ‘for the years that dehydration data was collected it was reported….?
Table 1: Why are % data not reported for dehydration?
Table 1: Caption would be clearer if the terms Number and Percent were used. The reader is not told what N refers to.
Section 3.1 states that 71 of 72 confirmed dehydration cases were included in the analysis (no explanation given), 72 cases are noted in Table 1.
Table 2: heading of first column: Suggestion: Annual rate per 100 workers.
Table 3: heading of first column: Suggestion: Annual rate per 1000 workers.
Figure 3: Caption, Line 257: per 1000 sugarcane harvesters (also need consistency of this term for Figures 1 and 2).
- Discussion
Lines 315-318: This is the first time Chronic Kidney Disease of Unknown Origin is mentioned. This could have been included in the Introduction as part of the rationale for the analysis of work-related dehydration.
Limitations: Do you see any limitations associated with self-reported data, use of just one weather station, or data for analysis of dehydration available for only two harvest seasons?
- References
Reference 3, citation appears to be incomplete
Reference 10, inconsistent use of commas and the symbol ‘&’, when compared with reference 26, for instance.
Author Response
Major comments
- This paper is interesting and a worthwhile contribution to the discussion of the impact of heat on workers, in the context of a warming climate.
Response: Thank you for this comment.
- The Introduction needs to make a clearer rationale for including analysis of the impact of WBGT on the risk of dehydration in this study. There is also limited data available for analysis of this issue. One option would be to include this in a separate study once more data has been collected. This is a very important issue too.
Response: We agree. We have updated the introduction section at line 55 to include a paragraph on why dehydration was an important outcome to include in the analysis.
- The limitations section of the discussion needs to address the issues of self-reported data (potential bias), use of a single weather station and limited data about dehydration.
Response: We have updated the discussion section at line 387 to reflect the issues of using recorded injury data, a single weather station, and the limited information on dehydration available for these analyses.
- I notice that there is little discussion about ethics considerations of this study, other than to say that the research had been deemed ‘non-human subject research’. Was any consent required from workers? Were they aware that data was being collected about them?
Response: Thank you for bringing this issue to our attention. This was an oversight on the author’s part. These data were previously collected by the agribusiness for business purposes. The proposed post hoc use of these de-identified data was in fact reviewed and approved by our Institutional Review Board. We have updated the manuscript at line 130 to indicate the IRB approval number for the evaluation of these data.
Minor comments
- Introduction
A more in-depth rationale for including an assessment of the relationship between WBGT and risk of heat-induced dehydration illness in the study aim is needed. This appears somewhat ‘out of the blue’ in the final paragraph of the introduction.
Response: Thank you. We have updated the introduction section at line 55 to include a paragraph on why dehydration was an important outcome to include in the analysis.
Line 31/32: is any more recent data available? Reference 1 is dated 2014. This should be mentioned.
Response: Unfortunately, the AR6 is not expected until 2021. We have updated the introduction sentence to mention that the data from the AR5 was an estimate as of 2014.
- Material and methods
- Sugarcane harvesting
Line 91/92: what does collaborating for this analysis mean? Did all these workers consent to providing their data for the research? This should be stated more clearly.
Response: We realize that collaborating can mean many things. In the instance of this manuscript, the only role of the agribusiness was to provide the de-identified data. We have updated the manuscript to state: “Each harvest there are approximately 4,000 male manual sugarcane cutters employed at the mill providing data for this analysis.” We have updated the ethics section at line 130 to more clearly state the IRB review.
Line 100: Were workers provided with clean water or did they have to walk some distance to access this? Was any time penalty associated with accessing the water and electrolytes? This some elaboration.
Response: The implementation of the hydration, rest, and shade is an important component. We have updated the manuscript at line 108 to provide more detail around this process. Water sources are chlorinated and regularly checked for coliforms and metals. Workers were educated about the merits of hydration, electrolytes, rest, and shade and encouraged, but not disciplined, for health behaviors. There are no time penalties.
- Data Sources
Line 120/121: Why was this research deemed ‘non-human subjects research’?
Response: As noted this was a misunderstanding on the author’s part. We have updated the manuscript at line 130 to include the IRB approval number for this evaluation. In the terminology of our IRB, if using an existing data set that was not collected for the purposes of research, the work is considered an evaluation not human subjects research. The awkward phrase is direct language that we are instructed to use by our IRB.
Line118: does this mean data was available for just November to April of each financial year? It would be good to state this.
Response: This is correct. We have updated the manuscript at line 128 to include: “Data for this analysis were available for November through April from the 2014-2015, 2015-2016, 2016-2017, and 2017-2018 harvests.”
2.4.1. Outcome variables
Line 138: What was the rationale for using information about dehydration starting with the 2016/2017 season, rather than the 2014/2015 season? Was the 2014/2015 and 2015/2016 data less complete? This needs some explanation.
Response: Prior to the 2016/2017 harvest dehydration was not recognized as a reportable incident by the agribusiness. We have updated the manuscript at line 150 to clarify this.
Line 142-144: Please explain the rationale for this.
Response: We originally made the decision to remove this observation because the extrapolation to rate per 1000 workers was based on only 4 workers being present that day. However, in addressing this reviewer’s comment we re-ran the analysis with the inclusion of this dehydration case. Inclusion did not change the conclusions. For the sake of transparency, we made the decision to include all available data points in this revised version.
- Results
Line 187: grammar, should be ‘there were’…
Response: We have updated this.
Line188: grammar, should be ‘there were’…
Response: We have updated this.
Line 193: needs rewording. Do you mean ‘for the years that dehydration data was collected it was reported….?
Response: We have updated the language at line 212 to improve clarity.
Table 1: Why are % data not reported for dehydration?
Response: We did not include dehydration in the total injury count. We have clarified this in the footnote of table 1.
Table 1: Caption would be clearer if the terms Number and Percent were used. The reader is not told what N refers to.
Response: We have updated the caption and footnote to table 1 to more accurately reflect what the data in the table describe.
Section 3.1 states that 71 of 72 confirmed dehydration cases were included in the analysis (no explanation given), 72 cases are noted in Table 1.
Response: We have resolved this by including the previously removed case in the analysis.
Table 2: heading of first column: Suggestion: Annual rate per 100 workers.
Response: Thank you for this suggestion, we have updated the heading to include workers.
Table 3: heading of first column: Suggestion: Annual rate per 1000 workers.
Response: Thank you for this suggestion, we have updated the heading to include workers.
Figure 3: Caption, Line 257: per 1000 sugarcane harvesters (also need consistency of this term for Figures 1 and 2).
Response: Thank you for noting this, we have updated the captions to consistently reflect that the rates are for sugarcane harvesters. We have updated the manuscript to be consistent with our terminology of sugarcane harvester.
- Discussion
Lines 315-318: This is the first time Chronic Kidney Disease of Unknown Origin is mentioned. This could have been included in the Introduction as part of the rationale for the analysis of work-related dehydration.
Response: Thank you for this suggestion. We have updated the introduction to include a paragraph around the importance of analyzing dehydration and have made mention of CKDu in the paragraph at line 55.
Limitations: Do you see any limitations associated with self-reported data, use of just one weather station, or data for analysis of dehydration available for only two harvest seasons?
Response: We have updated the limitations section to acknowledge these limitations by including the following language at line 387: “The reliance on worker reporting and employer recording of data regarding both incidence and nature of the injury data likely did not capture all injuries leading to potential biasing of our results. Dehydration data were collected over only two years and only for those individuals who recognized the signs and symptoms, thus reducing our ability to detect an effect of WBGT on dehydration incidents. Measurements of WBGT were calculated using data from a single weather station. While our selection was based on proximity to the majority of fields where the sugarcane harvesters were working, we were unable to geographically locate each injury site. The choice of a single weather station for WBGT exposure measurement likely led to misclassification bias. A sensitivity analysis conducted on a subset of the data demonstrated that our observed results were sensitive to the selection of weather station.”
- References
Reference 3, citation appears to be incomplete
Response: Thank you for noting this, the citation has been updated.
Reference 10, inconsistent use of commas and the symbol ‘&’, when compared with reference 26, for instance.
Response: Thank you for noting this, the citation has been updated.
Reviewer 2 Report
My review in brief:
An analysis of the risk of injuries in relation to workplace climatic conditions among heavy manual labourers in a tropical climate is indeed of considerable interest. I congratulate the authors to this effort, having modelled WBGT effect based on approximately 10,000 workers, each observed over a 140 days harvest.
My main concern about this paper is the lack of careful discussion about the validity of the outcomes. In total, 201 accidents were reported over the 4 year period. The description “Occupational injuries were reported by the worker to their supervisor overseeing field operations that day or the field nurse who was present who then logged the incident” in the Method section gives the impression that not only major accidents were recorded, but also “incidents”.
This description contrast markedly with i.e. the finding of 14 reported total injuries in 2017-18 over a harvest with approximately 2.85 million working hours (The observed 0.68 injuries/100 workers in 2017/18 can be compared with i.e. 2.7/100 reported injuries in 2018 in the US (overall private industry, https://www.bls.gov/news.release/pdf/osh.pdf), clearly showing a very selective reporting). Thus, a general expression of “occupational injuries” in the title as well as throughout the paper is not warranted.
The research team has a longstanding relation with the mill within a Total Worker Health approach, and have repeatedly visited the field sites. It should thus have been possible to get better understanding on the collection of data on injuries, and description of data in the database obtained from the company, enabling a critical discussion. Just to mention “underreporting” briefly mentioned in two sentences in the discussion is not enough.
I also miss a discussion on the implication of selecting one WBGT monitoring station. If there is more information available, a brief discussion on spatial variability and microclimates would have been of value. Also, a brief notion on the relation between mean and max daily temperature ( and also modelled, if data exist) would have been valuable.
Author Response
My review in brief:
An analysis of the risk of injuries in relation to workplace climatic conditions among heavy manual labourers in a tropical climate is indeed of considerable interest. I congratulate the authors to this effort, having modelled WBGT effect based on approximately 10,000 workers, each observed over a 140 days harvest.
Response: We thank this reviewer for their constructive review of this manuscript.
My main concern about this paper is the lack of careful discussion about the validity of the outcomes. In total, 201 accidents were reported over the 4 year period. The description “Occupational injuries were reported by the worker to their supervisor overseeing field operations that day or the field nurse who was present who then logged the incident” in the Method section gives the impression that not only major accidents were recorded, but also “incidents”.
This description contrast markedly with i.e. the finding of 14 reported total injuries in 2017-18 over a harvest with approximately 2.85 million working hours (The observed 0.68 injuries/100 workers in 2017/18 can be compared with i.e. 2.7/100 reported injuries in 2018 in the US (overall private industry, https://www.bls.gov/news.release/pdf/osh.pdf), clearly showing a very selective reporting). Thus, a general expression of “occupational injuries” in the title as well as throughout the paper is not warranted.
Response: Thank you for this comment and we agree. We have updated the title to include “recorded.” Throughout the manuscript we have clarified the recorded nature of the injuries.
The research team has a longstanding relation with the mill within a Total Worker Health approach, and have repeatedly visited the field sites. It should thus have been possible to get better understanding on the collection of data on injuries, and description of data in the database obtained from the company, enabling a critical discussion. Just to mention “underreporting” briefly mentioned in two sentences in the discussion is not enough.
Response: We agree that underreporting is a major issue and limitation within this particular workforce. We have reworked the discussion section to highlight this issue and provide more insight to the safety culture of the organization and potential reasons for underreporting or under recording within this population. This can be found in the paragraph at line 357.
I also miss a discussion on the implication of selecting one WBGT monitoring station. If there is more information available, a brief discussion on spatial variability and microclimates would have been of value. Also, a brief notion on the relation between mean and max daily temperature ( and also modelled, if data exist) would have been valuable.
Response: Thank you for noting this important limitation. We have updated the limitations section at line 391 to discuss the issues surrounding the selection of a single weather station for the analysis. To explore the issue of microclimates we have included a supplementary sensitivity analysis examining the effects of mean WBGT on injury rate during the 2015-2016 harvest for both the El Balsamo weather station (used in the primary analyses) and the Cengicaña weather station. Additionally, we have added in the results section under WBGT (line 234) that daily mean and daily max WBGT were highly correlated (r = 0.88). As a sensitivity analysis we re-ran our final total injury model with centered max WBGT and provide the results in a new supplemental table.
Reviewer 3 Report
The authors predict the risk of occupational injury in sugarcane agriculture workers in a specific location, arising from from the global rise in WB global temp.The authors through their analysis show that the rise in WGBT temp increases risk of occupational injury. The article is well written and the study design well conceived. However there are three major concern the author's have to address sufficiently
(1) Based on Table 1 and figure 1, the rate on injury seems to have reduced over the years, which is contradicting the observation that rise in WGBT alone leads to occupational injury. For instance in Figure 2, 2014-2015 WGBT and 2017-2018 WGBT ( x axes) are almost same, except 2017-2018 has on value at 34 WGBT which may be a outlier/ or random event. The cluster of spread of WGBT in these two chucks of years are more or less same, however 2015-2016, thre were more WGBT occurrence around 34 compared to rest of the years. This also gives some evidence that WGBT is not monotonically rising. Although in the discussion ?(line 326) the authors suggest that this trend with reduction in injury with year was because the employer took safety measures, the data suggests that there could be other reasons too. The authors have to elaborate on this in the discussion.
(2) Occupational injury and prevention in general is a multi faceted problem and so is the solution. The authors seems to have written the article in a very regression analysis data driven approach rather than a "system thinking" approach. For instance, such occupational injuries are also a fucntion of individual muscle strength, the tool design, individual health status, the quality of sleep before work, other extrinsic factors like shoe ( slip surface for falls) and psychosocial aspects, habits like ( smoke, alcohol), age of worker etc. The authors never talk about these things and these were not accounted for in the model. So without including these factors, in any form, but just claiming WBGT rise can increase risk is not a very solid approach. The authors have to add a section in the discussion about these so that the reader has the opportunity to see the problem as a while system aspect. And also add the lack of addition of these aspects to the regression model as potential limitation.
(3) Figure 1 X Axis must be "year"?
(4) Lines 311 and 312 states risk of injury was highest during first 2 weeks. Please update figure 3 to include the first 2 weeks as well.
(5) Conclusion states "Acknowledging that agricultural workers will be at an increased risk of occupational injury will allow businesses to plan and adapt by implementing and evaluating effective OSH programs to protect the health and safety of their workers." Any occupational injury initiatitve is a collective goal, and unless it is participatory with the workers itself the company alone cannot achieve that goal. I recommend the authors to rephrase this to create a holistic perspective, rather than singling it out as just the responsibility of the business.
All the best to the authors.
Author Response
The authors predict the risk of occupational injury in sugarcane agriculture workers in a specific location, arising from from the global rise in WB global temp.The authors through their analysis show that the rise in WGBT temp increases risk of occupational injury. The article is well written and the study design well conceived. However there are three major concern the author's have to address sufficiently
(1) Based on Table 1 and figure 1, the rate on injury seems to have reduced over the years, which is contradicting the observation that rise in WGBT alone leads to occupational injury. For instance in Figure 2, 2014-2015 WGBT and 2017-2018 WGBT ( x axes) are almost same, except 2017-2018 has on value at 34 WGBT which may be a outlier/ or random event. The cluster of spread of WGBT in these two chucks of years are more or less same, however 2015-2016, thre were more WGBT occurrence around 34 compared to rest of the years. This also gives some evidence that WGBT is not monotonically rising. Although in the discussion ?(line 326) the authors suggest that this trend with reduction in injury with year was because the employer took safety measures, the data suggests that there could be other reasons too. The authors have to elaborate on this in the discussion.
Response: We thank the reviewer for noting this. We initially left this observation out as we did not want to be criticized for overly attributing the reduction in injury to WBGT alone. We have now included this observation in the discussion at line 373.
(2) Occupational injury and prevention in general is a multi faceted problem and so is the solution. The authors seems to have written the article in a very regression analysis data driven approach rather than a "system thinking" approach. For instance, such occupational injuries are also a fucntion of individual muscle strength, the tool design, individual health status, the quality of sleep before work, other extrinsic factors like shoe ( slip surface for falls) and psychosocial aspects, habits like ( smoke, alcohol), age of worker etc. The authors never talk about these things and these were not accounted for in the model. So without including these factors, in any form, but just claiming WBGT rise can increase risk is not a very solid approach. The authors have to add a section in the discussion about these so that the reader has the opportunity to see the problem as a while system aspect. And also add the lack of addition of these aspects to the regression model as potential limitation.
Response: We thank you for this critical review and agree. We have reworked the discussion section to highlight the potential importance of accounting for individual vulnerabilities and have proposed the Total Worker Health framework to examine the issue holistically. Additionally, we have acknowledged our inability to account for individual risk factors as a potential limitation. This section of the discussion is found at line 376. The overall findings of our data evaluation produced a seminal observation, however we agree that there is a great need for future research to incorporate these other likely contributors.
(3) Figure 1 X Axis must be "year"?
Response: Thank you for bringing to our attention that the axis labeling was unclear. The data plotted along the x-axis is daily. We included the year to indicate the differing harvest seasons. We have clarified this in the caption to figure 1.
(4) Lines 311 and 312 states risk of injury was highest during first 2 weeks. Please update figure 3 to include the first 2 weeks as well.
Response: Thank you for this suggestion. We have updated Figure 3 to include the estimated line for the first 2 weeks as well.
(5) Conclusion states "Acknowledging that agricultural workers will be at an increased risk of occupational injury will allow businesses to plan and adapt by implementing and evaluating effective OSH programs to protect the health and safety of their workers." Any occupational injury initiatitve is a collective goal, and unless it is participatory with the workers itself the company alone cannot achieve that goal. I recommend the authors to rephrase this to create a holistic perspective, rather than singling it out as just the responsibility of the business.
Response: We agree with this review and have updated the language in the conclusion to state: “Acknowledging that agricultural workers will be at an increased risk of occupational injury will allow employees, employers, and regulatory agencies to plan and adapt by implementing and evaluating effective OSH programs to protect the health and safety of the agricultural workforce” at line 408.
All the best to the authors.
Response: Thank you.
Round 2
Reviewer 2 Report
The authors have now clarified that this paper is based on reported injuries, and also discuss obstacles for reporting from workers.
In a dangerous job, prone to minor and major injuries, the number of reports is far below what is found even in high-income countries. It would have been appropriate to check the character of reported injuries better. With only 111 cuts over 4 harvests in a workforce of 2700 workers engaged in manual cane cutting, my guess is that only major injuries were reported. The same is likely true for falls and slips. If so, this should also be clearly clarified in the discussion.
I do think the last year of data is peculiar - is that level of positive impact in any complex industry really possible to achieve in one year? Would the findings of the study be similar if this year is removed from the analysis?
A comment on a sentence in the discussion: "With the understanding that manual harvesting of crops must be conducted regardless of temperature, it is imperative to determine the excess risk that workers assume when they labor under conditions of higher temperatures". I recommend another word - the workers do not assume the risk. They experience it daily.
In summary, the paper has improved and can be accepted, although I still have recommendations for minor revision.
Reviewer 3 Report
Thank you addressing all the concerns.
Best wishes.